# The transcriptional legacy of developmental stochasticity

Sara Ballouz[1,5,9], Risa Karakida Kawaguchi [1,6,9], Maria T. Pena[2], Stephan Fischer [1,7], Megan Crow[1,8], Leon French [3], Frank M. Knight[4], Linda B. Adams [2] & Jesse Gillis [1,3] ✉

Genetic and environmental variation are key contributors during organism development, but the influence of minor perturbations or noise is difficult to assess. This study focuses on the stochastic variation in allele-specific expression that persists through cell divisions in the nine-banded armadillo (*Dasypus novemcinctus*). We investigated the blood transcriptome of five wild monozygotic quadruplets over time to explore the influence of developmental stochasticity on gene expression. We identify an enduring signal of autosomal allelic variability that distinguishes individuals within a quadruplet despite their genetic similarity. This stochastic allelic variation, akin to X-inactivation but broader, provides insight into non-genetic influences on phenotype. The presence of stochastically canalized allelic signatures represents a novel axis for characterizing organismal variability, complementing traditional approaches based on genetic and environmental factors. We also developed a model to explain the inconsistent penetrance associated with these stochastically canalized allelic expressions. By elucidating mechanisms underlying the persistence of allele-specific expression, we enhance understanding of development's role in shaping organismal diversity.

Cells in a fully developed organism share life histories traced back through their divisions, defining lineages. Epigenetic marks left on a single cell early in development can be inherited down through cell divisions, leaving shared features across cells, and barcoding their lineages[1,2]. The clearest example of this happening in nature is X-chromosome inactivation which is an epigenetic process that regulates gene dosage in females[3,4]. Occurring as a random coin-flip in each cell early in development, the status of inactivation is then stably inherited down cell lineages (via, e.g., DNA methylation)[5,6]. Unusually, this is an example where the lineage relationship between cells can be observed by eye, as in the calico cat, where color alleles are X-linked

and create obvious patterning[3]. But X-inactivation (XCI) is only one inherited epigenetic mark; cells likely have thousands, progressively defining cell-type as the cells move down Waddington landscapes in complex relationships creating stable developmental trajectories[7,8]. Further discovery and characterization of the shared marks, mechanisms, and impact of cell lineage relationships remain a central goal of modern biology.

Attempts to ascertain the existence of these permanent shared markings in previous work have mainly focused on the strongest events – monoallelism – in the simplest systems – cell lines – with mixed results[9–11]. Monoallelism is interesting because it could reflect

[1]Cold Spring Harbor Laboratory, Cold Spring Harbor, NY 11724, USA. [2]US Department of Health and Human Services, Health Resources and Services Administration, Healthcare System Bureau, National Hansen's Disease Program, Baton Rouge, LA 70803, USA. [3]Physiology Department and Donnelly Centre for Cellular and Biomolecular Research, University of Toronto, Toronto, ON, Canada. [4]University of the Ozarks, Clarksville, AR 72830, USA. [5]Present address: School of Computer Science and Engineering, Faculty of Engineering, University of New South Wales Sydney, Sydney, NSW, Australia. [6]Present address: Center for iPS Cell Research and Application, Kyoto University, Kyoto, Japan. [7]Present address: Institut Pasteur, Université Paris Cité, Bioinformatics and Biostatistics Hub, Paris F-75015, France. [8]Present address: Genentech, Inc., South San Francisco, CA, USA. [9]These authors contributed equally: Sara Ballouz, Risa Karakida Kawaguchi. ✉e-mail: jesse.gillis@utoronto.ca

regulatory noise from differentiation that is propagated forward epigenetically[9]. While some studies have reported this inherited effect in cell lines[12], more recent work assessing individual cells in tissues has suggested some effect but the exact degree of impact has been challenging to ascertain[13] with studies variously suggesting abundant monoallelism without lineage-dependency[10] and a major role for cell intrinsic noise such as bursting[14]. One possibility is that early lineage marks are broadly encoded but invisible since, unlike XCI, they are not aligned across chromosomes in a simple-to-observe way. If so, these marks will be confounded with the effect of eQTLs in studies of wild populations, or with genetic background in crosses of inbred strains.

One solution to the challenge of controlling for genetic background is to exploit identical twinning. Identical twins will frequently

have maternal and paternal alleles that are derived from the same genetic background while still being distinct enough to permit allelic analyses. While genetics can be well controlled in identical twin studies, environment is more of a challenge. This is particularly critical for functional genomics studies, which generally measure properties that are responsive to environmental variation. To ensure both genetics and environment are shared in an outbred organism, we turned to *Dasypus novemcinctus* (the nine-banded armadillo) which has a polyembryonic reproductive strategy, producing litters of identical quadruplets (quad) (Fig. 1a–c). The splitting of the blastocyst into 4 embryos is first observed after it implants in the uterine wall and forms the epiblast cell layer[15], but distinct cell lineages may have formed earlier. As armadillos are identical, any variant that has the same

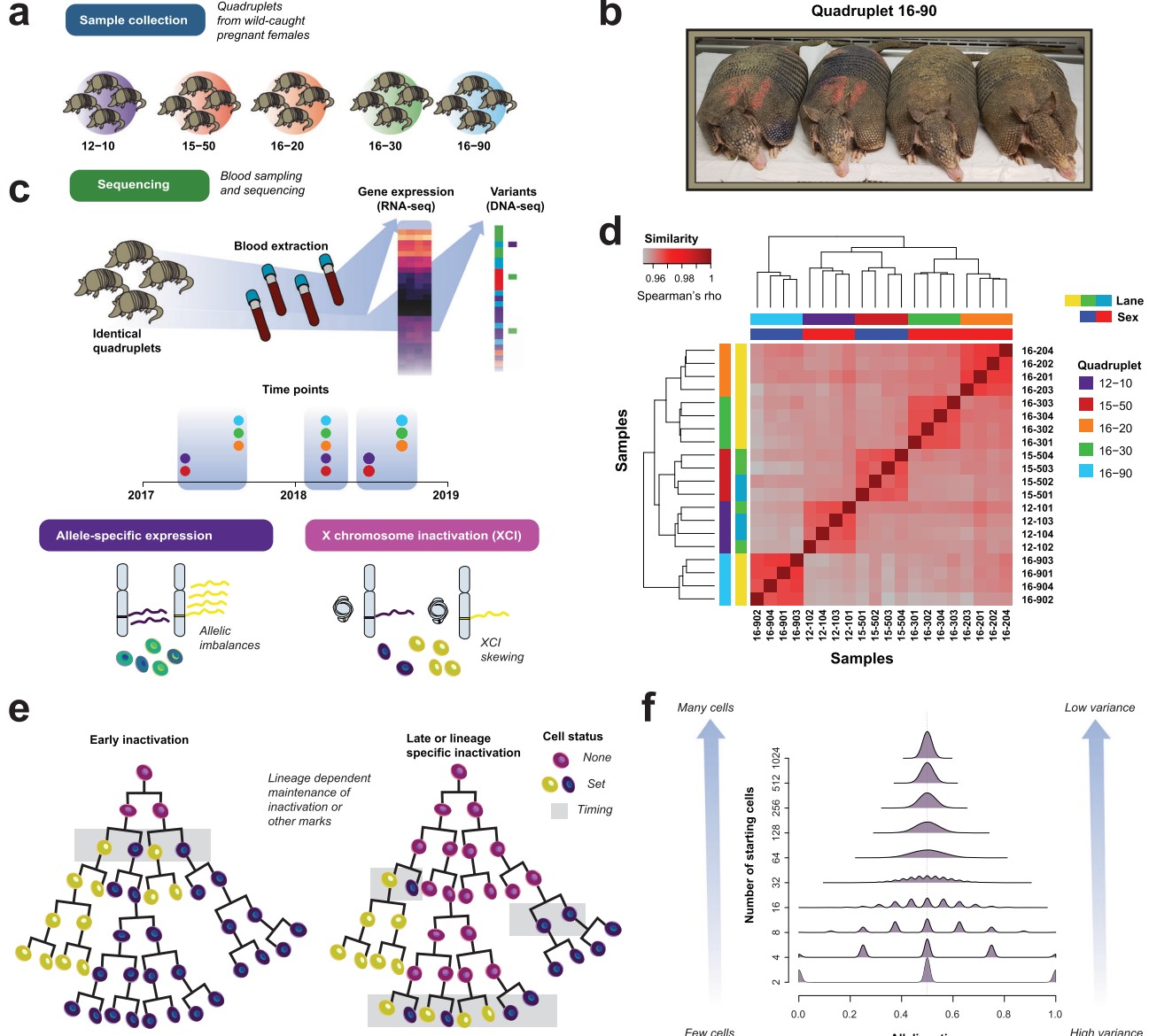

**Fig. 1 | Defining transcriptional identity. a** Samples were collected from 5 unrelated armadillo quadruplets. **b** An armadillo quadruplet. **c** Schematic of the study design (top): blood samples were collected and sequenced at 3 time points for each armadillo quadruplet. Within each quadruplet, we assess sibling identity from allelic expression (bottom). **d** Transcriptional similarity is higher between siblings than across armadillo quadruplets. Heatmap of sample-sample Spearman's correlations across all genes for the first time point. The leftmost row annotations and top column annotations indicate quadruplet identity with color, the lower column annotation marks sex (red = female, blue = male) and sequencing lanes for each sample are shown in the second-row annotation bar. **e** Developmental timing of epigenetic marks can be estimated by calculating the starting number of cells required to generate the range of allelic imbalances observed. **f** Densities of allelic ratios (x-axis) plotted against the number of starting cells (y-axis) illustrate that high variance in allelic ratios is associated with early inactivation timing (few cells). Source data are provided as a Source Data file.

impact on gene expression will not distinguish individuals, allowing us to systematically assess the organismal impact of autosomal epigenetic events. While armadillos are not a traditional model in genomics, they are the model system for the study of Hansen's disease (Leprosy) and a small number of colonies exist where they are reared in captivity.

In this work, we study allelic imbalances from blood samples collected at 3 time points over the course of 18 months from 5 wild-caught armadillo quadruplets (Supplementary Table 1, 20 armadillos in total, ages 1–6, 3 female quadruplets). Because each litter of armadillos shares a genotype and environment, they are not drivers of gene expression differences between individuals. While individuals within litters share an environment at any given time, it is not a fixed one, with one major source of variability being the infection of the armadillos with the leprosy bacterium toward the end of our study (as a primary purpose of the colony). While this would be a striking experimental design decision from first principles, we think it does little to diminish our environmental control (since all comparisons are internal) and, in fact, is likelier to ensure results are robust across typical large scale changes in environment rather than overfitting to a single environment.

In typical model systems, differences in gene expression are likely to arise from transient noise, environmental perturbation or distinct genetics. Absent these effects, and particularly in the case of allelic differences, epigenetic regulation, set independently of genotype, will appear as random allelic imbalances, showing a preference for one allele over the other on average. If this epigenetic decision is faithfully preserved down the lineage, the allelic imbalance may be copied even as the cell population rises, yielding a preference for one allele over the other on average, distinct across individuals. In essence, we exploit the fact that there will be some noise in the exact balance between two alleles at the moment the expression level is set within a lineage. If, as in XCI, this balance is inherited down the lineage it will be possible to observe permanent imbalances in the adult organism. In combination, these allelic imbalances progressively barcode trajectories across cell lineages as they move forward during development.

We first focus on the X-chromosome, determining the timing of XCI from the distribution of allelic ratios alone. Further, we show that variation in ratios across individuals within a quad predicts its stability over time. Building on this, we model the predictability of allele-specific expression (ASE) imbalances to estimate our power to resolve cell lineages barcoded on the autosomes. We find that for our read coverage and number of armadillos, we are powered to detect events arising as late as the 10,000 cell stage. We show that autosomal allelic ratios varying between individuals are also enriched for stability over time, consistent with an early developmental origin. We test the ability of these autosomal ratios to barcode individuals and find that, in combination, they define approximately a half X-chromosome worth of additional epigenetic signal in terms of their contribution to organismal transcriptional variability. We close by suggesting this is likely to have an important impact on disease variability by rendering otherwise haplosufficient genetic variation penetrant.

## Results

In order to better understand the range of expression variability within and across the quadruplets, we first assessed the similarity of the expression profiles. Similarity of the overall gene expression within and across quadruplets at a given time point is extremely strong (Fig. 1d, Supplementary Fig. 1). The minimum correlation obtained between gene expression profiles of any two armadillos is >0.95, high by most standards. However, the range within quadruplet sets is higher still, averaging approximately 0.99, leaving quadruplet sets easily seen on the heatmap of correlations. Our target, signatures of individuality or stochasticity within quadruplet sets, are consequently quite subtle, given this overall strong similarity between individual armadillos in overall profile. To establish features associated with their presence and

validate our general approach, we first turn to allelic ratios of the X-chromosome in the female quads, where epigenetic variability is clearest, as a biological ground-truth.

## Allelic distributions are barcoded by X-inactivation

Within mammals, one of the most prominent transcriptional features of epigenetic individuality is XCI in females[4]. During development, epigenetic marks (e.g., DNA methylation and histone modifications) are deposited, and then maintained along a cell's lineage (Fig. 1e)[6]. Allele specific expression, i.e., the expression of genes at the level of their variants, and allelic ratios – the fraction of expression attributed to each allele – allows us to assay the aggregate output of these epigenetic marks. As XCI is stochastic when it first occurs and then is maintained in cell lineages, it creates permanent variability between individuals. And because XCI ratios differ only by virtue of random sampling from within the original cell population, that population size directly defines the degree of variance observed (Fig. 1f)[16].

In our female samples, we can readily see variability in which X-chromosome is expressed (Fig. 2a). To obtain allelic ratios, we aligned RNA sequencing reads to quad-specific personalized genomes, identifying a total of 26,325 heterozygous SNPs across the 5 quads. On highly powered heterozygous SNPs from the X chromosome (>5 reads on each allele), the average folded allelic ratios ranged from 0.52 to 0.63. By plotting the distribution of these ratios across the 3 female quads (12 individuals), we can estimate XCI from an initial cell population of approximately 25 cells (Fig. 2b), plus or minus approximately 1 cell division. After adjusting for reference bias, our estimate remains in a range of 10–100 cells (Supplementary Fig. 2). This timing suggests that stochastic canalized variation between siblings is set in distinct cell lineages before separate armadillo embryos are observed (Fig. 2c). The X-chromosome offers a particularly strong opportunity to test whether factors other than lineage create extreme allelic distributions (across individuals sharing genetics). To test this, we calculate the significance of the variability in allelic ratios for each SNP on the X chromosome (departure from a binomial model with a shared allelic ratio, see Methods). Where allelic distributions are significantly variable, it is evidence that which allele is expressed is not independently chosen – even after controlling for genetics – across the sample of cells contributing to the aggregate expression. As we become more confident that an allelic signature arose early in development, we are able to observe that it is also likelier to be stable into the future (Fig. 2d).

Having validated that the timing and inheritance of XCI leaves a well-defined permanent impact in allelic signatures across individual SNPs or at the chromosomal level, we looked for a more general measure that allows SNPs to be combined, as on the X, but might generalize to autosomal signatures not necessarily shared on a whole-chromosome basis. This measure of individuality can be quantified by the degree to which knowing it permanently defines the individual relative to siblings. Our basic strategy is to compute confusion matrices for each SNP's allelic ratio across individuals within quads between time points (Supplementary Fig. 3, Methods). For canalized SNPs, the ordering of individuals is expected to be conserved across time points (diagonal confusion matrix). We then aggregate the confusion matrices across SNPs to predict identity (cross-validation across the 3 time points), reporting a score between 0 and 4, indicating how many individuals were correctly identified within the quadruplet sibling set. We then average prediction scores across all prediction sets (3 time point combinations, up to 5 quadruplets). Not surprisingly, the allelic imbalance ratios of the X genes are highly predictive of an individual within a quadruplet (Fig. 2e, average score = 2.56, 3 time points, 3 female quadruplets, $p = 5.9e\text{-}5$, Fig. 2f, $p = 0.001$). The high predictive scores of XCI indicates that individuals within a quadruplet set have distinct XCIs, at least with respect to the cell populations sampled in this study.

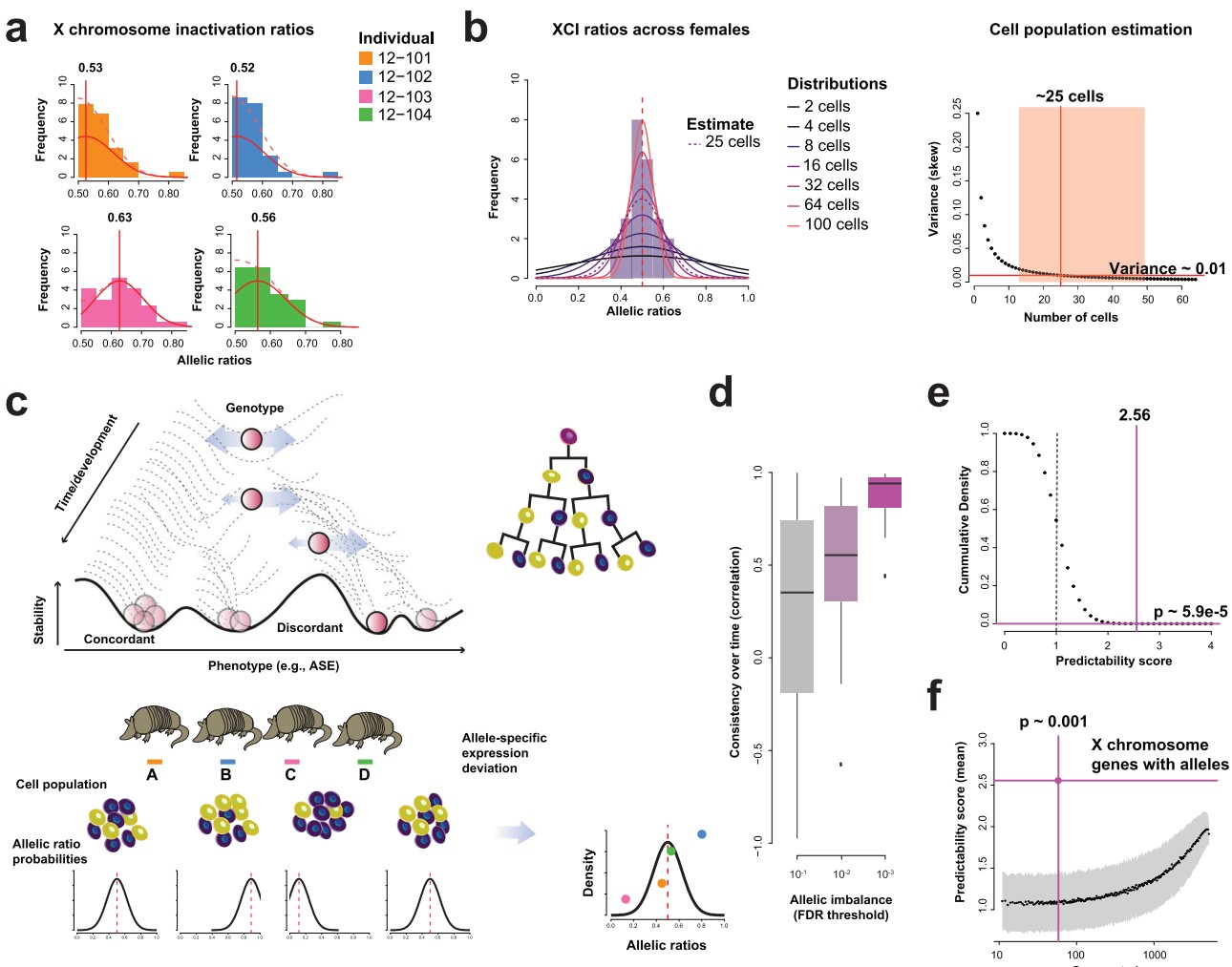

**Fig. 2 | X chromosome inactivation as a mark of individuality. a** Histograms of allelic ratios estimated across high-coverage SNPs on the X chromosome and maximum likelihood estimation of X-inactivation skews (red line = unfolded estimate, dotted line = folded estimate). X-inactivation estimates from RNA-seq data in quadruplet 12−10 shows variation in. **b** Histogram of X-inactivation skews estimated from high-coverage SNPs of the 12 female individuals and theoretical distributions from a binomial model (left). Expected skew variance depending on the number of cells at X-inactivation timing (right). The orange area indicates the range of cell numbers that are compatible with the observed skew variance. We estimate the number of cells where inactivation occurred to be around 25. **c** Gene expression is canalized, generating variable allelic imbalances inherited from early development. **d** Boxplots of allelic imbalance correlations across time points for X chromosome SNPs passing 3 FDR thresholds demonstrate that significant allelic imbalance

(excess variability across siblings in a quad) is associated with consistent allelic ratios over time (variability across 2387 X-SNPs from the 3 female quadruplets at 3 time points). The lower and upper hinges represent the first and third quartiles, respectively. The center bar indicates the median, while the whiskers extend 1.5 times the interquartile range above and below the boxes. **e** XCI strongly predicts identity within a quadruplet (one-sided empirical *p* value = 5.9e-5). Observed predictability score for sibling identity using X chromosome genes (purple vertical line, average across the 3 female quads and the 3 time points) against the theoretical distribution for random predictions (median score = 1, dashed line). **f** Plot of mean predictability scores (y-axis) versus gene set size (x-axis) for random gene sets and the X chromosome genes (highlighted with purple lines, one-sided empirical *p* value < 0.001). Source data are provided as a Source Data file.

## Individual SNPs exhibit canalized variability across individuals

While the cell-lineage allelic signature we see on the X chromosome in females is expected, extending it to autosomal genes has historically been a major challenge. The most common form of epigenetic regulation is imprinting, but as it is dependent on the parental genome, it is a poor marker of individuality. Instead, more subtle marks that control gene expression which are cell specific may be present, typically averaged away or only apparent under particular environmental stresses. In this scenario, alleles are imbalanced due to variation in the epigenetic regulation of these alleles, measured as consistent variation in allelic balance. Typically, allelic imbalance is attributed to the impact of a variant: either the variant within the gene has an effect on the stability of the mRNA, or an upstream SNV has cis-regulatory effects[17]. As armadillos are identical, any variant that has the same impact on gene expression will not distinguish individuals. Instead, epigenetic

regulation will appear as random allelic imbalance reflecting noise in the original assignment in an individual cell. At the organismal level, allelic imbalances reflect the compositional distribution of cell-lineages within an individual, where groups of cells are skewed in one direction for a set of genes.

To extend our analyses to SNPs on the autosomes, we used the same rationale as for allelic signatures left by X inactivation: if ASE is determined early and canalized down lineages, we expect the variance of allelic ratios to be high across individuals within a quad and to be conserved over time. Unlike XCI, autosomal SNPs affected by ASE are not known a priori. We therefore performed a power analysis to determine the minimum read coverage needed to identify individual SNPs with ASE from allelic imbalance alone. In order to capture potentially low coverage signal, we randomly sampled 1% of SNPs detected in our data and simulated ASE for these SNPs (individual-specific allelic ratios

following a binomial model, see Methods), which we refer to as ASE-SNPs. To measure our capacity to recover ASE-SNPs, we ordered SNPs by deviation from the null expectation that allelic ratios are identical across individuals within a quad (statistical significance from Chi-Squared test, see Methods), then measured the predictive power using the area under the Receiver-Operator Characteristic curve (AUROC). The AUROC measures the probability of ranking an ASE-SNP in front of a non-ASE-SNP, with an AUROC of 0.5 indicating random predictions, and 1 indicating perfect recovery of all ASE-SNPs.

The results suggest that ASE-SNPs can be readily identified at the read coverage observed in our data. ASE induced by early lineage events (≤64 cells) created high-variance allelic imbalance distributions that could be detected with read coverage as low as 12 reads (AUROC > 0.75, Fig. 3a, b). The predictive power remained remarkably high even for late lineage events (AUROC ~ 0.7, 64–10k cells, Fig. 3b), in particular for SNPs with high read counts (AUROC > 0.75 for SNPs with >95 reads), indicating that ASE-SNP predictions have the potential to barcode a wide range of developmental states. As genes affected by ASE-SNPs are expected to be similar over quads, power can be further increased by aggregating predictions over multiple quads. Indeed, the more quads are sampled, the higher the chance to observe a quad with extreme distributions of allelic imbalance. Accordingly, for ASE events induced at the 64-cell stage, the performance increase was particularly high when aggregating 5 quads instead of relying on a single quad (ΔAUROC ~ 0.1, Fig. 3c, d). Performance continued to increase with the number of quads, plateauing around 50 independent quads.

With these model results in hand, we next turned to the real ASE-SNPs in our data. As in XCI, our first evaluation of the SNPs showing excess allelism is to see if ratios are fixed over time. Canalized ASE events from early development are expected to have highly variable allelic ratios across individuals as they are propagated down the lineage, but also to be conserved over time within individuals. To test the overlap between these two properties, we predicted ASE-SNPs from the observed allelic ratios at time point 1 and asked whether they show evidence of being canalized at later time points. Concretely, we evaluate whether the ordering of ratios is preserved; e.g., whether the individual with the highest allelic ratio at time point 1 will still have the highest ratio at time points 2 and 3 (Methods, Fig. 3f). SNPs that show no significant allelic bias exhibit almost identically 0 mean correlation over time, highlighting the natural near-perfect control of allelic variability across fixed genotypes (Fig. 3e). In contrast, significant ASE-SNPs also showed significant excess correlation over time yielding a 30% enrichment over non-significant ASE-SNPs at high correlations (Fig. 3e). In essence, temporal consistency for a given individual is predicted by excess variability between individuals, i.e., when the observed variability is consistent with having arisen within an initially small population. That non-significant autosomal SNPs also show a distribution almost identically centered at a correlation of 0 over time is valuable in suggesting that the aggregation of signal across SNPs may be surprisingly straightforward.

### Combining canalized allelism yields strong signals of identity

In order to measure the aggregate impact of the observed stochastic variation in early cell lineage decisions, we combine allelic signatures across all SNPs. We use the same aggregation strategy as for the X chromosome, predicting identity from confusion matrices for each SNP's allelic ratio across individuals within quads between time points. As before, a SNP which shows significant and consistent allelic variability between individuals will predict those individuals at later time points (Fig. 3f). We then aggregate the confusion matrices to generate a barcoding of allelic ratios that characterizes individuals in aggregate and validate by the degree to which this is preserved over time.

Simply combining all genes (excluding those on the X chromosome) into a single confusion matrix yields highly significant predictability scores (Fig. 3g, average score = 2.2, p ~ 3.74e-5, 3 time points, 5

quads). In magnitude of impact, this is approximately equivalent to an additional half an X-chromosome worth of imbalance distributed across the genome of both females and males. Looking across all quadruplets, we can find an average of 700 genes exhibiting strong imbalances. This is much larger than an estimated germline de novo mutation rate of 19 to 21 per generation based on data from 36 mammals[18], consistent with our model that these allelic imbalances arise epigenetically, like XCI. Using these imbalanced genes as the feature set was predictive of individuality and gives approximately the same score as the genome in aggregate (Fig. 3h, p ~ 0.002 for a gene set of that size, permutation test), implying a significant fraction of these genes are epigenetically differentially canalized. Because of the relationship between cell population and variance, greater imbalance implies a smaller number of cells present when our signal originated, which can be explicitly evaluated as in our earlier model.

Assuming our signal is explained by 700 genes and that epigenetic marks are set at approximately the same time, we estimate that most of the observable variance is due to non-genetic marks set early in the development of the blood lineage, at around a few hundred cells (Fig. 3i). Given the nature of epigenetic regulation, this is plausible, as the reprogramming of the embryo and setting of marks occurs at these stages[19]. However, there are other population bottlenecks (e.g., progenitor pools) within specific cell lineages, so even an exact number of cells does not define an exact developmental stage without broader assessment across tissues to observe shared and distinct relationships. Functionally, the signature of the allelic imbalanced genes was unique to each sibling cohort of quadruplets, but did include enrichment for a common immune component (Fig. 3j), as expected by the cell lineages sampled within our experiment. However, the more prominent signals are related to signaling and enzymatic activities. Functionally, these are of interest in development as they ensure the switching on and off of programs that may result in phenotypic abnormalities if not controlled.

### Modeling haplosufficiency suggests canalized alleles may contribute substantially to disease

Having found major variability in the transcriptional endophenotype of our quadruplet sets, we next considered how this could directly affect the phenotype. Again, taking XCI as a conceptual model for the individual alleles on the autosome, one possibility is that skewing could contribute to the penetrance of otherwise haplosufficient genes. In essence, a gene whose transcriptional state is set early in development will exhibit highly variable allelic ratios, yielding a disease phenotype if the disease-associated allele dominates the lineage. Is this a plausible source of disease incidence within natural populations?

In order to model this effect, we consider a disease allele present in a population that becomes pathogenetic in an individual if it represents a certain fraction of the total gene expression (Fig. 4a). If the individual inherits two disease alleles, this trivially means they have insufficient levels of a functional copy of the gene and will exhibit the disease (but it will be more severe). Alternatively, a gene whose expression is set early in development will exhibit increased variance in the adult cell population that may yield haploinsufficiency.

Under this model, we calculated the probability that an observed disease phenotype arose via inheriting two disease variants as opposed to developmental stochasticity. As the disease variants become rare, the probability of inheriting two disease alleles becomes small much faster than the probability of inheriting only one and noise pushing it to be a high enough fraction of total expression to yield a disease phenotype. For example, at disease allele frequency 0.001, the disease is more likely to be caused by haploinsufficiency (probability > 0.5, Fig. 4b). For disease due to combinations of common variants, this model has essentially no explanatory power; i.e., such diseases will essentially never arise due to independent unlucky expression of deleterious variants (probability <0.5 at frequency 0.01 for allelic

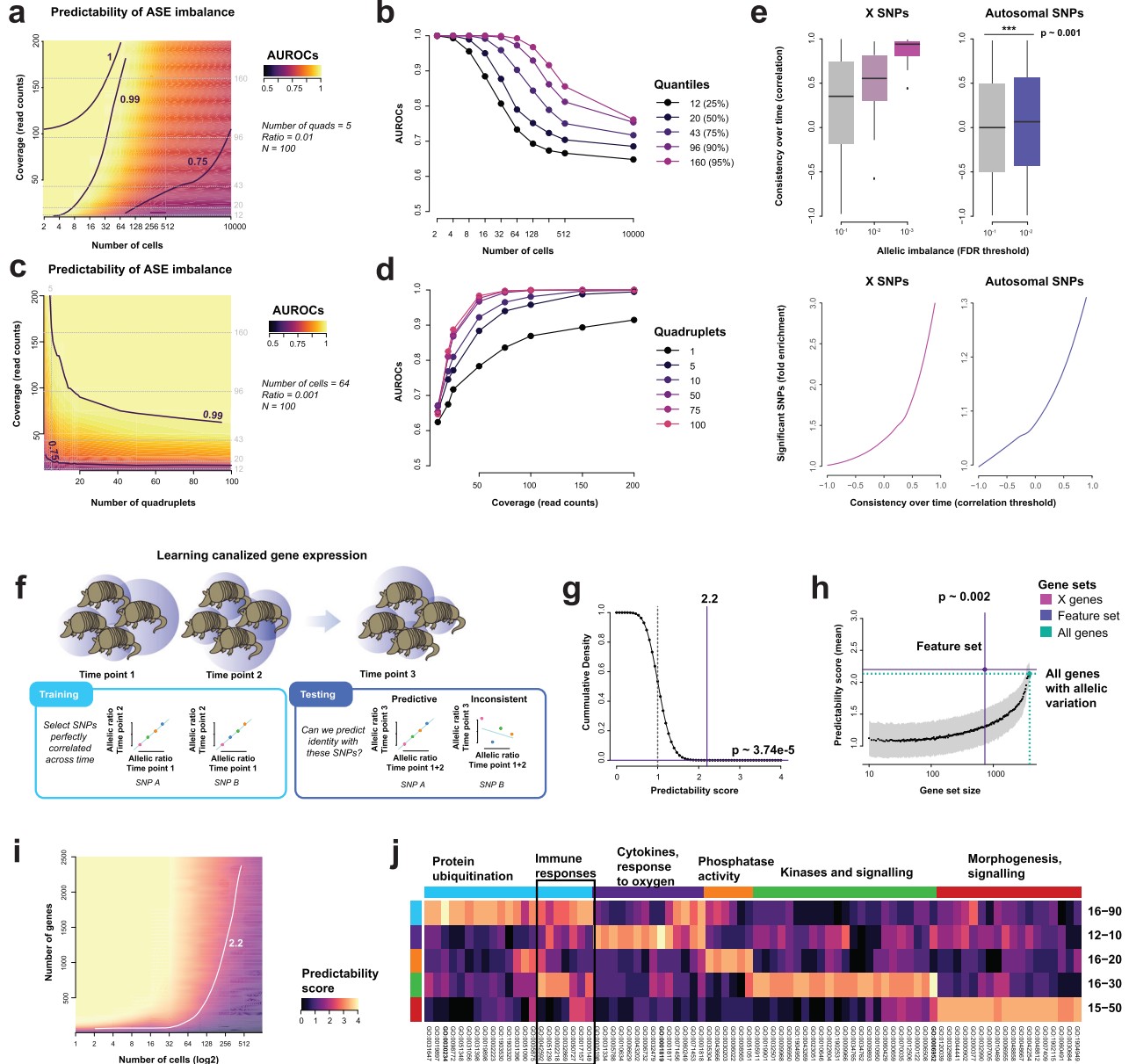

**Fig. 3 | Persistent allelic imbalance as a mark of individuality. a–d** Power analysis measuring ability to recover imbalanced SNPs from allelic imbalance based on SNP coverage, number of quads in which the SNP is measured, fraction of SNPs affected by true allelic imbalance (Ratio). Performance is measured across $N = 100$ independent simulations. **a** Ability to recover imbalanced SNPs depending on SNP coverage (read counts, y-axis) and timing of inactivation (in number of cells, y-axis). **b** Same data as A, focusing on 5 levels of coverage (marked with dashed gray lines in A and C). **c** Ability to recover imbalanced SNPs depending on SNP coverage and the number of quadruplets. **d** Same data as (**c**), focusing on 6 levels of quadruplet number. **e** Boxplots of allelic imbalance correlations across time points for X and autosomal SNPs passing 3 FDR thresholds show that SNPs with significant allelic imbalance have consistent allelic ratios over time (variability across 703,870 autosomal SNPs from 5 armadillo quadruplets at 3 time points, 2387 X-SNPs from 3 female armadillo quadruplets at 3 time points, one-sided Mann–Whitney test, $p < 0.001$). The lower and upper hinges represent the first and third quartiles, respectively. The center bar indicates the median, while the whiskers extend 1.5 times the interquartile range above and below the boxes. **f** Predicting identity from allelic imbalance. We select SNPs with perfectly correlated allelic ratios across two

time points (feature set, likely to be canalized), then make identity predictions on the remaining time point. **g** On aggregate, genes with allelic imbalances map to identity. The predictability score (x-axis) for sibling identity using all genes (solid vertical line, average across the 5 armadillo quads and the 3 time points) is significantly higher than for random predictions (dashed line, median score = 1, one-sided empirical $p$ value - 3.74e-5). **h** Predictability scores (y-axis) and gene set sizes (x-axis, log scale) for the feature set (solid purple lines) and all genes (dotted blue lines) shown relative to predictability scores obtained using random gene sets (black line and gray band). The score for the feature set was significantly higher than random sets (one-sided empirical $p$ value - 0.002). **i** Expected predictability score depending on the number of cells (x-axis, log scale) at which genes are canalized and the number of canalized genes (y-axis). We find that our signal is explained by 500–700 genes and time events to a few hundred cells present when epigenetic marks were set. **j** Predictability score for functional gene sets across the 5 armadillo quads. High predictability scores are explained by signaling and molecular functions unique to each quad. Source data are provided as a Source Data file.

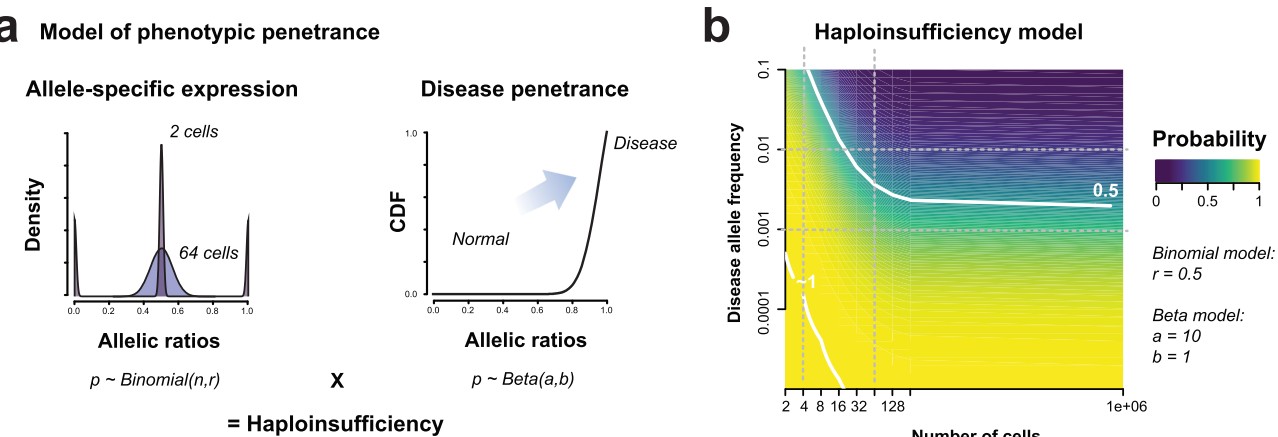

**Fig. 4 | Canalized genes may contribute to disease. a** Model where haploinsufficiency arises from the canalization of extreme allelic ratios, leading to disease penetrance. The allelic ratio of the disease-causing allele is given by a binomial model parameterized by the number of cells at which expression is canalized. Heterozygous individuals with extreme allelic ratios of the disease allele have an increasingly higher probability of developing the disease (beta distribution). **b** Probability that a disease is caused by extreme allelic ratios rather than homozygosity for the disease allele, depending on the disease allele frequency (y-axis, log scale) and the timing of canalization (in log-scaled number of cells, x-axis). Source data are provided as a Source Data file.

imbalance determined after the 16 cell stage, Fig. 4b). In contrast, this model suggests a very large role for stochasticity in the impact of rare variants, particularly in disorders that are known to arise during development. Such studies have observed a sharp enrichment for loss of function variants combined with haploinsufficiency, but the degree of phenotypic heterogeneity for a given genotype is very high, with significance for variants largely arising across genes rather than for individual ones[20]. This is consistent with a model in which the early developmental stochasticity for allele choice is a major determinant of final phenotypic penetrance of a heterozygous individual.

## Discussion

In human studies, expression variability is most simply thought of as being due to some combination of environmental and genetic factors, leaving "noise" purely as a nuisance term. More complex models may include epigenetic associations of phenotypic and behavioral interactions[21], but canalized noise is still rarely considered (although see, for example:[22,23]). However, random effects provide an extremely useful record of shared history, a fact exploited in lineage analyses of all types[24–27]. To discover the permanent imprint of noise on cells in the organism, our study design exploited the shared environment of alleles within cells. In armadillos, this cellular control is particularly strong since the genetic background is perfectly shared. Thus, extrinsic noise (i.e., from outside the cell) is shared by the two alleles, leaving variability in expression of alleles potentially driven by intrinsic noise[28,29]. Because this intrinsic noise is inherited down cell lineages, it leaves a permanent mark shared across many cells. While other study designs have, for the same reason, also focused on engineering or exploiting cases where allelic output can be carefully measured[10,30,31], armadillos offer unusual robustness and replicability by sampling across identical sets of independent outbred genetic backgrounds. That they are mammals also provides a useful reference for the epigenetic signals we uncovered in the form of XCI, where the results of intrinsic variability are clear. Our key finding in armadillos is that similar allelic imbalances exist on the autosomes and are stably preserved and substantial in aggregate effect, totaling approximately half an X-chromosome worth of signal in both males and females in this species.

Our study has a number of limitations due to our use of a non-model organism and our interest in subtle inter-individual effects. Key limitations include our use of peripheral blood mononuclear cells (PBMC) as our sample source, reference genome quality in the

armadillo, the number quadruplet sets and variability between quadruplet sets. PBMCs offered the advantage of sampling over time without sacrificing the animals. This is key in studying natural heterogeneity over developmental time in contrast to pseudo-development in a functionally clonal group (e.g., isogenic mice sacrificed at different stages). Of course, the use of PBMCs potentially limits the generality of the results. However, in humans, we have shown that XCI ratios are shared across tissues[32]. Although the number of quadruplets used in our study was relatively small, our experimental design, which involved sampling at different time points within each individual of a quad, provided an unusually well-structured framework in which many genetic and environmental effects are controlled. Our experimental design was enabled by our use of armadillos, but leveraging this non-model organism does create challenges with respect to genome annotation. We resolved this by sequencing the individual genomes and assembling the X-chromosome; this yields much higher quality data but at a much higher expense. In contrast to the environmental control within quadruplet sets, we did have variability between quads due to the alternate use of the armadillo colony to study Hansen's disease (at later ages). Since virtually all our analyses were within quads, this has little impact on our results, other than showing they are robust to shared environmental changes which would normally occur. However, this may explain one of our key findings, which was limited functional overlap in the ASE signatures across quads.

Lack of overlap could reflect fundamental biological differences between the quadruplet litters, or it could reflect our ability to detect distinct ASE. We think of this as being a question of whether the Waddington landscape is somewhat distinct between litters. If each litter has a subtly distinct landscape (partly due to their genome and partly due to their environment including, e.g., pathogen exposure), it is very easy to imagine minor perturbations creating unique signals in each litter. On the other hand, if all armadillos share a landscape (for the purpose of our analysis), lack of overlap is more surprising. However, it could be explained by a broad signature such as our data otherwise supports, and technical limitations being the key driver of which subset of that broad signature we detect. Technical limitations in this context could include anything affecting our power to detect canalized allelic signatures such as quite plausibly different factors such as read depth, gene expression level, the allelic ratio itself, or even reference bias. We suspect that our results reflect a combination of both models.

Considering the implications of our work more broadly, variability in human phenotype is the product of genetic and environmental

contributions, along with a complex interplay between the two[31]. While genomic data has permitted valuable progress in our understanding of both heritable and non-heritable phenotypic variation, this progress has been more piecemeal in sources of non-heritable variation. All studies of genetic or environmental influences on phenotype are affected by this unexplained, non-heritable variability or 'noise'[33]. One possibility is that 'noise' can be partitioned into well-defined categories of its own, based on underlying mechanisms. Development has long been thought to be a potential driver of unexplained phenotypic variability[34] it is a time when small initial changes can permanently propagate forward to large later effect (Fig. 2c). While programmatic variability in development has received particular attention[35,36], our works shows that early random effects could be a major source of phenotypic variance, particularly in the context of disease. In order to measure this developmental stochasticity, tight environmental and genetic control are necessary to minimize external drivers of variability, while outbred genetics are necessary to maximize the likely functional implications of observed variability. Although precise developmental timings may vary between mammals, the armadillo provides a valuable model system, with the closest parallel being outbred diversity crosses in mice.

The degree of skewing toward one chromosome over the other has been researched intensively[37–40], and importantly it has been linked to disease, where female carriers of X-linked disorders can have differential disease penetrance as a function of skewing[41–43]. Historically, researchers have looked to causal mechanisms for this effect[41,44,45], although our recent work suggests that the observed distributions are perfectly consistent with the expected number of cells in humans[32]. Cell selection is less of a concern in the case of the armadillo data since, if driven by either genes or environment, it is controlled within the experimental design. We suggest our approach is potentially powerful as a lineage tracing technique[32], even though it relies on statistical barcoding across populations of cells, rather than barcoding of individual cells (which is inherently noisy). This diminishes the impact of, e.g., transcriptional bursting, which has led to substantial controversy about the presence of monoallelism[10,46]. Our results reconcile previous observations by showing that while such permanent allelic effects exist, they are far weaker and more graded than individual cellular measures would easily reveal. However, they represent an important marker of lineage decisions within the organism since they appear to mark early events in development that are permanently inherited, marking classical epigenetic events.

## Methods

### Armadillo collection and samples

This study complied with the NHDP Institutional Animal Care and Use Committee (IACUC) guidelines under the approved protocol "Leprosy research support and maintenance of an armadillo colony" (A-102). Five sets of armadillo quadruplets (20 armadillos in total) were used in this study(Supplementary Fig. 4, Supplementary Table 1). Sex was considered in the study design. Pregnant females were captured using long-handled nets at night from the wild in 2012, 2015 and 2016. Capture of the pregnant females was done during the spring to avoid collecting females who were nursing young, but were potentially pregnant. The animals were retrieved from the nets and placed in kennels for immediate transport to the holding facility at the University of the Ozarks, Clarksville, AR. The pregnant females were kept in outdoor pens that had burrows where they gave birth to the quadruplets. The babies were kept with the mothers until they were observed foraging at about 6–10 weeks postnatal age. After separation from the mothers, the animals were housed together in semi-outdoors pens (rubber covered concrete floor under a roof). Most litters in the semi-outdoor pens shared the pen with another litter, either from this study or a separate one (2 litters per pen). All adults and young over 49 days postnatal age (pna) were fed a mix of dry dog and cat chicken-and-rice chow moistened with water—in an approximate ratio by volume of 1:1:2. Adults were provided 0.75–1.5 cups (indoor-outdoor) of moistened chow once a day during the gestation and 1.25–1.75 cups per day during known or suspected lactation. Animals housed in outdoor enclosures were able to forage as well. Occasionally, a raw egg and earthworms were provided in addition to the chow. Litters were fed replacement formula of reconstituted Esbilac puppy replacement formula until old enough[47]. After 35 days pna, the diet was gradually transitioned to that of adult by 49–56 days pna. The wild-caught females were administered 0.15 ml Ivermectin SC, and Exceed antibiotic if they showed any wounds or abscesses. Adults were dewormed every 6–8 weeks with Panacure on chow for three consecutive days, or with 0.2 ml Ivermectin on food. The babies were treated once with Panacure on chow for three consecutive days.

At four to five months of age, the animals were delivered to the National Hansen's Disease Program (NHDP) facility in Baton Rouge, LA where siblings were placed in pairs in modified rabbit cages[48]. They were fed the same dry food as that given at the Arkansas facility. After a period of adaption of approximately one year, the animals were treated with Penicillin (1.0 mL) and dewormed with Ivermectin (0.1 mL) and Praziquantel (0.4 mL). Prednisone (10 mg/mL) was also given at this time.

### Armadillo time course analysis and Hansen's disease

Blood samples were collected at three time points per quadruplet staggered over the course of a year, starting from March 2017 until August 2018 (Supplementary Table 1). The pilot study consisted of two sets of quadruplets (12–10 and 15–50), and then later blood was obtained from quadruplets 16–20, 16–30 and 16–90. The names of the armadillo quadruplets were chosen according to the year of capture of the pregnant mother (e.g., 16–XX for 2016). The analysis of the overall transcriptome and the top 1000 highly variable genes (Fig. 1d, Supplementary Fig. 1) suggests that all quads are roughly equally distant from each other (no evidence for any clear grouping or driver of variability such as age or SNP overlap shown in Supplementary Fig. 6). During the course of the year, the original two of the five sets of armadillo quadruplets were infected intravenously in the saphenous vein with $1 \times 10^9$ *Mycobacterium leprae* derived from athymic nude mice[49] – both after the first time point was collected. Blood was collected at different time points throughout the course of disease and at 18–24 months post-infection, the animals were humanely sacrificed when they developed heavy *M. leprae* dissemination with severe hypochromic microcytic anemia. The bacteria will locate in the bone marrow and the animals will eventually succumb to secondary complications of persistent bacteremia if not sacrificed[50].

### Armadillo RNA-sequencing

Blood was collected from the subclavian vein in BD Vacutainer Glass Mononuclear Cell Preparation (CPT) tubes (Fisher, USA), and PBMC were isolated following standard protocols[51]. Blood collection was performed under general anesthesia using Ketamine HCL (10 mg/kg) and Dexdomitor (0.1 mg/kg). All animals were screened for leprosy and their health (CBC and blood chemistry) evaluated at tri-monthly blood screenings. RNA was extracted from the PBMC using an automated Maxwell 16 Instrument (Promega) and a Total RNA purification kit (Promega). Library preparation was done with a poly(A) selection kit (KAPA mRNA HyperPrep) to enrich for mRNAs. Multiplexed paired end sequencing (PE76) was done using an Illumina NextSeq500 on multiple flow cells. We blocked for lane batch effects by splitting the quadruplet samples into pairs and ran two pairs of each set per flow cell (Supplementary Table 2). We downloaded the armadillo genome (Das-Nov3) from *Ensembl* (v95)[52], and generated an index file for use within STAR[53]. We mapped reads with STAR and standardized counts to counts per million (CPM) by summing the counts and dividing by 1e6 (Supplementary Fig. 5 and Supplementary Table 3).

## Armadillo DNA-sequencing

DNA was extracted from blood collected according to standard protocols. We sequenced each quadruplet together to obtain their identical genome sequence. We pooled DNA from all four individuals of a quadruplet, except in the case of quadruplet 16−30 where we could not get enough DNA from individual 16−301 (Supplementary Table 4). An average of 2.3 µg of DNA per quadruplet were sent for whole genome sequencing at the New York Genome Centre (NYGC). Library preparation was Illumina TruSeq Nano DNA, 450 bp. Sequencing was done on the NovaSeq with $2 \times 150$ bp. Coverage depth was 30X. Reads filtered on quality and were aligned to the DasNov3.0 genome from NCBI using BWA[54]. Variants were called from the BAM files using the GATK Unified Genotyper[55] following best practices for DNA variant calling[56] (Supplementary Fig. 6).

## Armadillo personal genome generation

We used g2gtools (unpublished, https://github.com/churchill-lab/g2gtools v0.2.0) to generate a personal quadruplet genome for each quadruplet set. We first created VCI files of the SNPs and INDELs using the g2gtools vcf2vci with the −pass and −quality tags. This is an indexed version of the VCF file required by g2gtools. Homozygous (alternate) SNPs and INDELs that passed quality control were kept. SNPs were incorporated into reference genome FASTA file using the g2gtools patch command. INDELs were then incorporated into the patched genome with the g3gtools transform command. A chain file was generated using the g2gtools vcf2chain command. We updated the genome annotation file (liftover) using the new genome with the g2gtools convert command. As the genome of the armadillo is not assembled beyond a large number of scaffolds, the patches and transformations were done per scaffold. Once completed, we concatenated all the scaffold FASTA files back into one. With these five personal genomes, we generated individual STAR indices. Using samtools (v.1.9), we generated index files for the new genomes, and dictionary files with picard from GATK (v3.6.0)[55].

## Armadillo personal genome mapping and allele specific expression analysis

Following quality control, we mapped reads from each quadruplet to their personal genome with STAR (v2.7)[53] (Supplementary Table 5). The resulting bam files were then run through GATK's v3 best practices pipeline[56] to filter for quality alignments. Briefly, the pipeline involves adding read groups, marking duplicates, and then splitting and trimming based on CIGAR. A WIG file was then built using the count command in IGVTools (v2.3.80)[57]. We then generated a VCF file with the heterozygous and homozygous (alternate) SNPs for each quadruplet. This VCF file was converted to a BED file, and then liftover to update the coordinates to the personal genome. This was then converted back to a VCF file. The SNPs (VCF) and counts (WIG) were then overlapped to obtain allele specific counts. Once again these were all performed on individual scaffolds, and recombined at the end of the analysis, which allowed for parallelization of the pipeline.

## Defining the armadillo X-chromosome

As the genome of the armadillo is unassembled, we constructed the X-chromosome by identifying which scaffolds were most syntenic to mammalian X-chromosomes. As the X-chromosome has high synteny between mammalian species (e.g., mouse and humans 95%, Supplementary Fig. 7), we used alignments of armadillo scaffolds to the X-chromosome of both human and mouse. We used the UCSC[58] chain/liftover files between the armadillo genome and the human (hg38) and mouse (mm10) genomes. We extracted the scaffolds from these files that align to the respective Xs of the species. There were over a million human alignments (1,231,264) to around 2 K armadillo scaffolds (Supplementary Fig. 8). The largest and most overlapping to the human X was scaffold JH573670.1, but holds no annotated human X

homologs. To the mouse, there were less than a million alignments (873,607) to around 1.6 K armadillo scaffolds. The largest is once again scaffold JH573670.1. We included smaller scaffolds with a high overlap (90% alignment) with the human and mouse X as the remaining potential X scaffolds. We consider these scaffolds to represent most of the X-chromosome of the armadillo. As a final X identifier, we located an *XIST* homolog which is not annotated in the current annotation. Using the human *XIST* sequence (NC_000023.11), we performed a BLAST[59] search on the armadillo genome. Of the 16 hits that were to annotated armadillo genes, we then performed a reverse BLAST on the human genome to find the reciprocal top hits. The two genes (ENSDNOG00000033080 and ENSDNOG00000047775) match to two *XIST* exons, and both these genes belong on the same armadillo scaffold (JH583104.1) and are within a few 100 Kbp (Supplementary Table 6). These two genes were also hits using the mouse *Xist* (NC_000086.7, Supplementary Table 7). Using these genes as placeholders, we could derive the rest of *XIST* from the read pileups (Supplementary Fig. 9). The locus is JH583104.1:145,010-175,550.

## Building functional annotation sets for the armadillo

Currently, no gene functional annotations exist for the armadillo. We used the gene annotations from *Ensembl*[52] to generate a gene ID map between human and armadillo homologs. From the total of 33,374 coding and non-coding genes and transcripts annotated for the armadillo, there are 13,492 human homologs. In close parallel to the GO[60] annotation project's own process, we built an armadillo ontology using human gene-GO annotations[61]. Within our current mapping, on average, each armadillo gene belongs to ~85 GO groups, and each GO group has on average ~56 genes.

## X-chromosome inactivation analysis and cell number estimates

For every female armadillo, we estimated the X inactivation ratios of genes with alleles. For this, we took the variants called on the X scaffolds. For each gene, we combined the three timepoints by adding the count data. Since we do not have phasing information but wished to summarize the allelic ratios to a single gene, we took the most powered SNP (that with the most reads) as the representative SNP and calculated the allelic ratio as

$$ASE_g = r_g / c_g \tag{1}$$

where $ASE_g$ is the allelic ratio for gene $g$, $r_g$ is the number of reads mapping to the reference allele, and $c_g$ is the total read count. The gene ratios $ASE_g$ were then used to estimate the X skewing ratio for the individual female armadillos. For each individual $i$, we fitted a folded normal to the ratios $\{ASE_g\}_{g \in chrX}$ and used the maximum log likelihood estimate to obtain the estimated overall folded skew $f_i$. Finally, we took the variance of the estimated unfolded skew values $\{1 - f_i, f_i\}_{i \in females}$ to estimate the number of cells (N) in the original starting pool[62]. The formula for the variance of a binomial distribution was used. Since we assume that the probability of a cell inactivating either X is 0.5, $p = q = 0.5$ so the formula becomes:

$$N = \frac{pq}{Variance} \tag{2}$$

$$N = \frac{1}{4 Variance} \tag{3}$$

## Detecting allelic imbalance

To detect SNPs with significant allelic imbalance, we modeled the ASE of the specific SNP $g$ from one of the alleles using the binomial

distribution as follows

$$Binom\left(k, c_g, p_g\right) = \binom{c_g}{k} p_g^k \left(1 - p_g^k\right)^{c_g - k} \qquad (4)$$

where $k$ is the amount of sequencing reads that contain the reference allele for $g$, $c_g$ is a total read count overlapping the genomic location of $g$, and $p_g$ is the probability that the expressed mRNA is transcribed from one of the alleles with $g$. Because all individuals within the quadruplet are genetically identical, we assume the null model in which no epigenetic canalization occurred and $p_g$ is shared within the quadruplet. $p_g$ for each quadruplet was estimated in two different ways; one is the ratio of ASE by pooling all sequencing reads from the quadruplet and another is the average of the ASE ratios computed for each individual. Because $p_g$ values estimated in those ways were highly correlated (Pearson correlation coefficient = 0.989), we only show the results for $p_g$ estimated by the pooling method.

To test the deviation from the null model that all four individuals have an identical ASE ratio $p_g$ (no canalization at any stages), we used Fisher's method to combine the $p$-values computed for each individual as follows:

$$-2 \sum_{i=1}^{4} \ln P\left(k \le k_i, c_g, p_g\right) \qquad (5)$$

where $k_i$ represents the observed ASE for the $i$th individual. While Fisher's method is known to follow a $\chi^2$ distribution, the observed distribution was largely different from the expected null distribution. Several reasons may explain this phenomenon, such as reference biases or stochastic sequencing errors. To obtain a more conservative result, we sampled ASE ratios from the null binomial distributions (with common parameter $p_g$, but preserving individual coverage) for 10,000 times, generating a null distribution of Fisher statistics from which we deduced an empirical $p$-value. To test the time-invariance of ASE ratios within the quadruplet, we computed the Pearson and Spearman correlation of a quad's ASE ratios between the first and later time points (Fig. 3f).

## Measuring identity

As a test for individuality, we developed a machine-learning method that tests for the relative consistency of expression across individuals from an armadillo quadruplet across time. This is equivalent to identifying differentially expressed genes, but rather than looking between two conditions or two individuals, it is across four. The idea here is that differentially expressed genes in this way are indicators of identity. We first select a feature set of genes based on correlations between two time points. For each gene, we calculate the Spearman rank correlation between the values across a quadruplet for one time point and a second time point. If the rank ordering is consistent (i.e., the correlation is 1), then this gene is selected as a feature gene. We then test for consistency in the third time point. As the first two time points are perfectly correlated, these genes form the training set, and the left out time point is the test set. A gene scoring matrix (4 by 4) is built per gene by comparing the ordering of the test and training data. Each individual gives a score of 1 to the test data individual it thinks it is (i.e., which rank it matches), and a 0 otherwise. We then sum all the feature gene scoring matrices to produce an aggregate scoring matrix. Then, in a winner takes all strategy, we calculate a score which represents the number of armadillos that correctly predict themselves. The final score is between 0 and 4, with 4 as perfect predictability i.e., each armadillo correctly identifies its future (or past) self. We repeat this three times, using the first and second time points as training, the first and third, and finally the second and third, and then testing in the left out time point. We average this across time points to get quadruplet specific scores, and also across all to get a final overall score for the analysis. We

calculate an analytic $p$-value for this score by convolution of the expected distributions. We calculate an empirical $p$-value by repeating the learning task on randomly selected genes.

## Minimum requirements to detect allelic imbalance

To determine the requirements for detecting SNPs with canalized allelic imbalance, we performed an in silico assessment of the SNP discovery with a wide variety of experimental conditions. We used five parameters as controlled variables; coverage of total read counts for each SNP of each individual (coverage), ratio of canalized SNPs (ratio), statistical replicates to reduce the randomness of AUROCs (N), number of armadillo quadruplets, and number of cells at the timing of canalization (cell). In our simulations, each SNP is assigned to be either canalized or randomly sampled from both alleles. For canalized SNPs, we first sampled the background (canalized) ASE level for each individual from a binomial distribution with $p = 0.5$ and $n = cell$. We then sampled haplotype specific reads from a binomial distribution with $p$ as determined in the previous step and $n = coverage$. Next, for each quad, we computed a $p$-value by extracting the two individuals with the minimum and maximum ASE ratios, then testing for equal ASE ratios using the chi-squared test (and Fisher's exact test if the minimum read count is 5 or fewer). For non-canalized SNPs, we sampled $p$-values directly from the uniform distribution on the [0,1] interval. The $p$-values were then converted into False Discovery Rates using the Benjamini and Hochberg procedure[63]. The significance of the ASE bias for each SNP was judged according to the minimum adjusted $p$-values across multiple quads. Finally, we summarized performance as an AUROC by framing results in a binary classification setting, asking how well the minimum adjusted $p$-values were able to predict the ground truth SNP status (canalized/random).

## Empirical models to estimate genes and lineage

From the identity analysis, we estimate the number of genes that could drive the signal through a series of empirical models. In our first model, we simulate our allelic identity experiment. We took the underlying allelic expression data and added a proportion of variance to a fraction of the genes. We then calculated the average identity performance. We then convert the variance of the underlying data to a number of cells estimate by extending the analysis from the X inactivation estimate. To summarize, we assume that autosomal genes that display allelic imbalances are under regulatory control and are being expressed either monoallelically (some cells express one or the other), or differentially (one allele is expressed at a higher or lower amount). We also assume that this is persistent across time, such that once a cell is committed to expressing an allele, its lineage will continue to express this allele at a similar or equal amount. We also assume that this choice is random in a pool of cells at the same time. With these in mind, we can estimate the number of cells and the fraction of the genome that gives rise to the performance observed.

## Haplosufficiency modeling

To model the relationships between hetero- and homozygous mutations and disease appearance, we used the combination of a binomial distribution and a cumulative beta distribution. Let $\mu$ be the probability of the occurrence of a disease-related SNP across the population. We assume random mating in the parental generation so that each allele is inherited independently with probability $\mu$, such that the fraction of homozygous and heterozygous individuals is defined as $\mu^2$ and $2\mu(1 - \mu)$, respectively. In our model, the individuals with homozygous alleles are assumed to always show the disease phenotype. On the other hand, the phenotype for heterozygous individuals depends on their ASE level, which is randomly drawn from a binomial distribution with $p = 0.5$ and a fixed population size $c$ representing the number of cells at the timing of X chromosome inactivation or any canalized regulation event. To model disease penetrance, we used the cumulative beta distribution

with parameters $\alpha = 4$ and $\beta = 1$, corresponding to increasing likelihood to develop the disease for higher ASE levels (Fig.4A). Note that we set $Beta(0, \alpha, \beta) = 0$ to be consistent with the case that no disease phenotype is shown in the absence of transcripts from the disease-causing allele. Finally, the probability of showing a disease phenotype for a heterozygous individual is obtained by

$$P(Disease, c, p | Heterozygous) = \int_0^c Binom(x, c, p) \int_0^q Beta(y, \alpha, \beta) dy\, dx \quad (6)$$

where $x$ corresponds to the number of cells expressing the disease allele and $q(= x/c)$ is the ASE level. We computed an approximated probability by computing the above sum for each $q$ from 0 to 1 with the step 0.000001 (using the average of $Binom(qc, c, p)$ and $Binom((q - 0.000001)c, c, p)$). Considering the probability of occurrence of the hetero- and homozygous genotypes, the ratio of disease individuals across the population is obtained as $P(Disease, c, \mu) = \mu^2 + 2\mu(1 - \mu)P(Disease, c, p | Heterozygous)$.

### Reporting summary
Further information on research design is available in the Nature Portfolio Reporting Summary linked to this article.

## Data availability
All data are available in the main text, source data, or the supplementary materials. The accession numbers for the sequencing datasets reported in this paper have been deposited in Gene Expression Omnibus (GEO) under GSE141951 and Sequence Read Archive (SRA) under SRP233269. Source data are provided with this paper.

## Code availability
Source code is available on GitHub and has been archived by Zenodo at https://doi.org/10.5281/zenodo.8433151.

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

## Acknowledgements

We thank Tony Zador, Paul Pavlidis, Peter Koo and Dan Levy for helpful discussion. We thank Amanda Withnell for animal care. This research was supported by the National Institutes of Health R01LM012736 and R01MH113005 (J.G.) and by the National Institute of Allergy and Infectious Diseases Interagency Agreement IAA 15006-004 (M.T.P. and L.B.A.). The content is solely the responsibility of the authors and does not necessarily represent the official views of the National Institutes of Health. Armadillo blood for the study was obtained from the National Hansen's Disease Program.

## Author contributions

S.B., R.K.K. and JG performed the analyses. J.G. designed the study. F.M.K. provided the armadillos. M.T.P. and L.B.A. maintained the cohort and collected the samples. J.G. interpreted the data and wrote the paper with assistance from S.B., R.K.K., S.F., L.F. and M.C..

## Competing interests

L.F. owns shares in and has received consulting fees from Quince Therapeutics (formerly Cortexyme Inc.). L.F. has additionally received consulting fees from PeopleBio Co., Keystone Bio, and a separate undisclosed company, for which confidentiality agreements prevent disclosure, and these arrangements are entirely independent of this work. The remaining authors report no competing interests.
