## [Peer Review File · Nature Communications]

The transcriptional legacy of developmental stochasticityREVIEWER COMMENTS

Reviewer #1 (Remarks to the Author):

The article entitled "The transcriptional legacy of developmental stochasticity" aims to compare the allele-specific expression imbalance in PBMC of the nine-banded armadillo. This species is unique as it shows a reproduction strategy that permits to identify non-genetic and non-environmental influences on the phenotype. It can therefore be used to investigate the developmental stochasticity in the expression of alleles. If this developmental stochasticity is well known regarding X-inactivation, it's much more complicate to assess for autosomes. However, it can be a significant source of organismal variability, usually hidden in what is called "noise", but that can be important in adaptation/evolution as well as to understand disease appearance.

The highest originality of this paper is based on the choice of the model species. The nine-banded armadillo produces monozygotic quadruplets that consequently share the same genotype and that are in the same controlled environment during its development. The remaining phenotypic variability among individuals from the same quad must therefore be the result of stochasticity, among which ASE is an important possible mechanism. This choice is great and is perfectly justified to assess ASE in mammals and its role in developmental stochasticity. Their main assumption is that predictability of ASE imbalance is an evidence of early epigenetic regulation resulting in canalization of allelic ratios. In that situation, autosomal allelic ratios varying between individuals are enriched for stability over time. To test that, they assessed ASE from blood sampled at 3 timepoints in each individual armadillo among five different quads and applied a machine learning workflow to construct a co-expression network.

They assumed that ASE is persistent across time, such that "once a cell is committed to expressing an allele, its lineage will continue to express this allele at a similar or equal amount". Their results showed that the range of expression variability within and across the quad is very stable at a given timepoint and that the signature of individuality within quad is therefore subtle. Looking across all quads, they found an average of 700 genes showing strong imbalances. Interestingly, the signature of the allelic imbalanced genes was specific to each quad, while being enriched for a common immune component. They conclude that "permanent allelic effects exist but that they are weaker and more graded than individual cellular measures would easily reveal".

They developed a very solid and sound bioinformatic workflow all along the study, which is well explained in the Methods section. Beside classical RNAseq analyses, they adapted different tools to specifically test their hypotheses, such as g2gtools to build a reference genome for each quad, and machine learning to estimate the predictability within each individual over the time. All the data and codes are available on their github account (<https://github.com/sarbal/ayotochtli>) which permits reproducibility.

I nevertheless have a few concerns that the authors should address, and that are explained below. It is not clear why two out of five quads were injected by *Mycobacterium leprae*. They explained it in the Methods but did not mentioned it in the results or discussion section. It is not obvious that this infection was necessary to address the main question and it added a possible confounding factor in the analysis. I would like to see more explanation/justification. Moreover, the number of analysed quadruplets is rather low (5). They effectively calculated a posteriori power to their analysis but stronger conclusions could have been reached with a higher number of replicates. The fact that different signature of allelic imbalanced genes was reported within each quadruplet is fascinating and open the door to new hypotheses regarding the role of stochasticity in phenotypic variation. However, it is not well discussed in the article. What are their hypotheses to explain such differences ? Are these differences random ? Or would they find the same allelic imbalanced genes if they repeated the experiment ? Moreover, the fact that these genes were enriched for an immune component are possibly a consequence of the cell types used in this study (PBMC). They should have discussed the importance of this choice of PBMC in their results.

Reviewer #2 (Remarks to the Author):

The study examines longitudinal gene expression variability in 5 wild armadillo quadruplets. This is a unique and very elegant study design providing important novel information on stochastic variation in gene expression. The design may generalize, or serve as a model system, to e.g. human monozygotic sibships. My comments are mainly aimed at clarifying the methodology used.

General comment: It is assumed that "XCI ratios differ only by virtue of random sampling from within the original cell population, that population size directly defines the degree of variance observed"

**What about technical variability, tissue (in this case whole blood), and selective differential survival of white blood cell populations carrying a specific X-inactivation pattern? Could the authors comment on how important they think these factors might be and if/how their results are affected or unaffected by this?

In the abstract, introduction and early results section, information is lacking on sample size (total number of quadruplets and number of males/females), tissue and age at which gene expression is measured.

Abstract: please clarify this sentence: "Greater temporal stability within an individual is predicted by variance across individuals":

** it is not clear how stability can be predicted by (cross-sectional?) variation between individuals. Also, in the context of the paper: does across individuals refer to both between and within family variation? (the expression across individuals also occurs elsewhere in the text; please define if this refers to variation between or within families, or both).

Introduction: would it be possible to give some more background information on this unique way of poly-embryonic reproduction in (this species of) armadillo? Is anything known about the moment of splitting, for example? (hours, days?).

"More recent work assessing individual cells in tissues have suggested some effect but the exact degree of impact has been challenging to ascertain..."

**I found this sentence a bit vague; what exactly is it that single cell studies have shown and what is the challenge that remains?

Results: As a general comment – I could not understand the results section and figures without a prior thorough read of the methods section. The authors may want to consider improving the readability of the paper by including a bit more information about their methods in the results section and figure legends. As an example: the second line of results refers to stability of overall gene expression as shown in figure 1 D. Figure 1D shows a cluster plot for one time point (the legend reads: "(D) Transcriptional similarity is higher between siblings than across armadillo quadruplets. Heatmap of sample-sample correlations for the first time point with example sample-sample scatterplots"). Thus, stability does not refer to variation that is consistent over time? The legend of figure 1D mentions with example sample-sample scatterplots, but I could not find the example scatterplots in the figure?

The numbering of families (2x beginning with 12 or 15 and the remaining 3 with 16): does this have a meaning? E.g. does 16 refer to 3 times the same mother?

Figure 1

- The correlations between overall gene expression profiles are large between all samples; I suppose that this might be driven by the many transcripts with an expression level at the extremes (hardly expressed at all versus abundantly expressed) – though it would be nice to show an example of individual level data (a scatterplot). It is not clear to me if the correlations are computed based on all transcripts or on the most variable subset? If it is based on all detected transcripts, have the authors considered computing the correlations for genome-wide expression profiles after standardization (i.e. Z-score) of the expression levels? Perhaps this might lead to a better discrimination of quadruplets, in terms of the amount of within-quad variation.

- Figure 1D shows "lane", could the legend explain what it represents (I did find the information in the methods section, but it would be nice to also include this information in the legend).

"Because this occurs after armadillo splitting,"

**Could the authors provide references/further details for this statement? At which stage (i.e. how many cells) does the splitting occur? (see also question above about introduction).

Not incorrect but difficult to read: defining a ratio as a difference (line98)

Figure 2.

*The graphics are fantastic, but the legend is too slim for me to understand what is shown exactly in each figure. Could the legend be expanded a little bit to describe more precisely what is shown in each panel?

Could the authors comment on the extent to which their results might be influenced by post-splitting de novo (DNA sequence) mutations? Is anything known about the de novo mutation rate in armadillos (or could this be discussed in the light of observations in other species, i.e., human identical twins)?

Line 242: this model assumes random mating in the parental generation?

Discussion: the opening sentence ("In human studies, expression variability is typically thought of as being due to some combination of environmental and genetic factors, leaving —noise purely as a nuisance term") does not do justice to the alternative explanations that have been offered, e.g. a third source that may consist of nonlinear epigenetic processes that can create variability at all phenotypical-somatic and behavioral-levels). I recognize that this statement is refined on the next page.

Methods: Is it possible that/if the authors have assessed if the different quadruplets are related to each other (i.e. either because their mothers are related or because their fathers are related), and if the authors might want to discuss this option in the light of the high overall between-quadruplet correlation of gene expression profiles and potential further impacts on results.

Most litters in the semi-outdoor pens shared the pen with another litter (2 litters per pen).

**statement sounds a bit odd to me, given that there were only 5 litters in total. Would it make sense to say that there were 2 pens with 2 litters and one pen with 1 litter?

At four to five months of age, the animals were delivered to the National Hansen's Disease Program (NHDP) facility in Baton Rouge, LA where they were placed in pairs in modified rabbit cages

** were they placed in pairs from the same litter?

During the course of the year, the original two of the five sets of armadillo quadruplets were infected intravenously in the saphenous vein with 1×10^9 Mycobacterium leprae

**I assume that this was done as part of the aim of another study. Could this be clarified and could the authors comment if this might have influenced results of the current study?

"Finally, we took the variance of the extremal estimated skew values to estimate the number of cells"

*Please define "extremal" (this term has not been introduced yet)

**Could the authors comment how the lack of a pre-existing armadillo genome assembly and gene functional annotations for the armadillo might have affected their results?

The X-inactivation analysis is not entirely clear to me. Does it combine information across all genes? And does it only use heterozygous genotypes? Is it based on the same information for all quadruplets (or is it also possible that different transcript variants are used to interrogate X-inactivation in quadruplet 1 versus quadruplet 2?)

Please note that the GEO accession number was missing – perhaps this was not available yet.

"The accession number for the sequencing datasets reported in this paper have been deposited in GEO under XXX"

Reviewer #3 (Remarks to the Author):

This is a laudible attempt to identify persistent allele-specific expression through the life of an animal. The mechanism could explain differences in penetrance of heterozygous rare variants. This is also modelled in the paper. A weakness of the paper is that the paper is purely based on RNA-seq data, and has not assessed DNA methylation levels (through genome-wide or targeted approaches).

Main criticisms that need to be addressed:

-It is important to stress from the start that RNA sequencing was done on PBMCs so that the stochastic variation observed is the variation observed in blood cells and may not reflect that of other tissues. Moreover, the estimate of the precursor pool of cells (25 plus or minus one cell division for X-inactivation) is the precursor pool of blood cells, and should not be confused with the number of cells present in the embryo at the time of X-inactivation.

-The authors should discuss the relation to published work in discordant monozygotic twins, e.g. a study on combined allele-specific methylation and gene expression, e.g.:
<https://www.nature.com/articles/s41380-021-01126-w> ;
<https://www.ncbi.nlm.nih.gov/pmc/articles/PMC6887657/>

-The legends to the figures should contain much more detail. The authors should indicate what is plotted, based on how many SNPs and samples, what are the assumptions in the model etc.

-The abstract needs to be rewritten to better explain the main points of the paper to a general audience. For example: the abstract starts with genetic and environmental variation. This is not what the paper is about. The paper is about the stochastic variation in allele-specific expression that persists through mitotic cell divisions. The term "developmental stochasticity" in the second sentence can still mean anything and the reader does not automatically associate this with gene expression. Make clear that this is about imbalanced expression from the two chromosomes. Also the implications for non- or variable penetrance and how this was modelled in the paper can be explained much more clearly.

Minor comments:

-Make more clear that (if understood correctly) the quads are monozygotic and not heterozygotic, so are genetically identical? The value of the use of an animal model with four monozygotic quads can become clear in the abstract already.

-Even when using personal genomes, reference bias in the alignment may still persist. Describe the procedures to check for the absence of reference bias.

-The Methods section "X-chromosome inactivation analysis and cell number estimates" would benefit from more formula annotations. Currently, descriptions like "These gene ratios", "extremal estimated skew values" are difficult to interpret.

Reviewer #1

The article entitled "The transcriptional legacy of developmental stochasticity" aims to compare the allele-specific expression imbalance in PBMC of the nine-banded armadillo. This species is unique as it shows a reproduction strategy that permits to identify non-genetic and non-environmental influences on the phenotype. It can therefore be used to investigate the developmental stochasticity in the expression of alleles. If this developmental stochasticity is well known regarding X-inactivation, it's much more complicate to assess for autosomes. However, it can be a significant source of organismal variability, usually hidden in what is called "noise", but that can be important in adaptation/evolution as well as to understand disease appearance.

The highest originality of this paper is based on the choice of the model species. The nine-banded armadillo produces monozygotic quadruplets that consequently share the same genotype and that are in the same controlled environment during its development. The remaining phenotypic variability among individuals from the same quad must therefore be the result of stochasticity, among which ASE is an important possible mechanism. This choice is great and is perfectly justified to assess ASE in mammals and its role in developmental stochasticity. Their main assumption is that predictability of ASE imbalance is an evidence of early epigenetic regulation resulting in canalization of allelic ratios. In that situation, autosomal allelic ratios varying between individuals are enriched for stability over time. To test that, they assessed ASE from blood sampled at 3 timepoints in each individual armadillo among five different quads and applied a machine learning workflow to construct a co-expression network.

They assumed that ASE is persistent across time, such that "once a cell is committed to expressing an allele, its lineage will continue to express this allele at a similar or equal amount". Their results showed that the range of expression variability within and across the quad is very stable at a given timepoint and that the signature of individuality within quad is therefore subtle. Looking across all quads, they found an average of 700 genes showing strong imbalances. Interestingly, the signature of the allelic imbalanced genes was specific to each quad, while being enriched for a common immune component. They conclude that "permanent allelic effects exist but that they are weaker and more graded than individual cellular measures would easily reveal".

They developed a very solid and sound bioinformatic workflow all along the study, which is well explained in the Methods section. Beside classical RNAseq analyses, they adapted different tools to specifically test their hypotheses, such as g2gtools to build a reference genome for each quad, and machine learning to estimate the predictability within each individual over the time. All the data and codes are available on their github account (<https://github.com/sarbal/ayotochtli>) which permits reproducibility.

I nevertheless have a few concerns that the authors should address, and that are explained below.

It is not clear why two out of five quads were injected by *Mycobacterium leprae*. They explained it in the Methods but did not mentioned it in the results or discussion section. It is not obvious that this infection was necessary to address the main question and it added a possible confounding factor in the analysis. I would like to see more explanation/justification. Moreover,

the number of analysed quadruplets is rather low (5). They effectively calculated a posteriori power to their analysis but stronger conclusions could have been reached with a higher number of replicates.

We thank the reviewer for their comments. We have expanded our introduction section to mention and explain the infection of the armadillos. Infection was not necessary to address our main question but was done to advance separate research on Hansen's disease. The armadillo colony is only available to us because the US Department of Health and Human Services uses it to study Mycobacterium leprae. We do not consider this to be a confounding factor because all comparisons are performed within quads. Thus, we think of it as sampling from wild population variability, but holding it constant within quads. This helps ensure results are robust and not simply present in armadillos that experience static environments. We now discuss this in the introduction:

"While individuals within litters share an environment at any given time, it was not a fixed one, with one major source of variability being the infection of the armadillos with the leprosy bacterium toward the end of our study (as a primary purpose of the colony). While this would be a striking experimental design decision from first principles, we think it does little to diminish our environmental control (since all comparisons are internal) and, in fact, is likelier to ensure results are robust across typical large scale changes in environment rather than overfitting to a single environment. "

We agree that profiling a larger number of quads could have strengthened our conclusions. However, the total number of samples is reasonably substantial (60), and our experimental design allowed us to control for within-quad genetic and environmental factors, thereby boosting the power to detect subtle changes. We now include a discussion of this and related limitations:

"Although the number of quadruplets used in our study was relatively small, our experimental design, which involved sampling at different time points within each individual of a quad, provided an unusually well-structured framework in which many genetic and environmental effects are controlled."

The fact that different signature of allelic imbalanced genes was reported within each quadruplet is fascinating and open the door to new hypotheses regarding the role of stochasticity in phenotypic variation. However, it is not well discussed in the article. What are their hypotheses to explain such differences ? Are these differences random ? Or would they find the same allelic imbalanced genes if they repeated the experiment ? Moreover, the fact that these genes were enriched for an immune component are possibly a consequence of the cell types used in this study (PBMC). They should have discussed the importance of this choice of PBMC in their results.

Our hypothesis is that there is substantial shared allelic variability that is canalized, but various other bottlenecks filter our ability to observe it, yielding low overlaps between quads. We have added a new paragraph in the Discussion section that describes two models that may explain the different signatures:

"Lack of overlap could reflect fundamental biological differences between the quadruplet litters, or it could reflect our ability to detect distinct ASE. We think of this as being a question of whether the Waddington landscape is somewhat distinct between litters. If each litter has a subtly distinct landscape (partly due to their genome and partly due to their environment

including, e.g., pathogen exposure), it is very easy to imagine minor perturbations creating unique signals in each litter. On the other hand, if all armadillos share a landscape (for the purpose of our analysis), lack of overlap is more surprising. However, it could be explained by a broad signature such as our data otherwise supports, and technical limitations being the key driver of which subset of that broad signature we detect. Technical limitations in this context could include anything affecting our power to detect canalized allelic signatures such as quite plausibly different factors such as read depth, gene expression level, the allelic ratio itself, or even reference bias. We suspect that our results reflect a combination of both models.”

In regards to the cell types used, we have also added the following text that describes the strengths and weaknesses of using PBMC:

“PBMCs offered the advantage of sampling over time without sacrificing the animals. This is key in studying natural heterogeneity over developmental time in contrast to pseudo-development in a functionally clonal group (e.g., isogenic mice sacrificed at different stages). Of course, the use of PBMCs potentially limits the generality of the results.”

Reviewer #2

The study examines longitudinal gene expression variability in 5 wild armadillo quadruplets. This is a unique and very elegant study design providing important novel information on stochastic variation in gene expression. The design may generalize, or serve as a model system, to e.g. human monozygotic sibships. My comments are mainly aimed at clarifying the methodology used.

General comment: It is assumed that “XCI ratios differ only by virtue of random sampling from within the original cell population, that population size directly defines the degree of variance observed”. What about technical variability, tissue (in this case whole blood), and selective differential survival of white blood cell populations carrying a specific X-inactivation pattern? Could the authors comment on how important they think these factors might be and if/how their results are affected or unaffected by this?

We thank the reviewer for their remarks. We agree that it is important to comment on technical variability and have added two paragraphs to the Discussion section (second and third). These paragraphs encompass the potential effects of reference bias, PBMC, the number of quadruplets, and other factors. In regard to our results, we also discuss how some of these factors may explain limited functional overlap in the ASE signatures.

In the abstract, introduction and early results section, information is lacking on sample size (total number of quadruplets and number of males/females), tissue and age at which gene expression is measured.

We have added information about samples across the different sections:

Abstract: “We investigated the transcriptome of blood samples from five wild monozygotic quadruplets over time“

Introduction: “we study allelic imbalances from blood samples collected at 3 time points over the course of 18 months from 5 wild-caught armadillo quadruplets (Table S1, 20 armadillos in total, ages 1 to 6, 3 female quadruplets).”

In the Results section, we now state the number of quads and individuals throughout so the readers are aware of our sample size.

Abstract: please clarify this sentence: “Greater temporal stability within an individual is predicted by variance across individuals”: it is not clear how stability can be predicted by (cross-sectional?) variation between individuals. Also, in the context of the paper: does across individuals refer to both between and within family variation? (the expression across individuals also occurs elsewhere in the text; please define if this refers to variation between or within families, or both).

We have rewritten our abstract and have removed the quoted sentence. Guided by your comment, we now summarize our results as: “Our findings reveal a remarkable and enduring signal of allelic variability along the autosomes, which distinguishes individuals within a quadruplet despite their genetic similarity.” Compared to the original phrasing, we sought to convey the intuition that these SNPs are incompatible with a null model where allelic variability is reset over time. Variation between individuals is stable over time.

We agree that “across individuals” is an ambiguous expression. In the paper, it refers to variability between individuals within quads. This is now stated explicitly throughout the paper.

Introduction: would it be possible to give some more background information on this unique way of poly-embryonic reproduction in (this species of) armadillo? Is anything known about the moment of splitting, for example? (hours, days?).

We have added information about when the separate embryos are observed in the Introduction section. Armadillos have relatively unusual reproductive strategy more generally, including delayed implantation. We have added the following text:

“The splitting of the blastocyst into 4 embryos is first observed after it implants in the uterine wall and forms the epiblast cell layer [15], but distinct cell lineages may have formed earlier.”

“More recent work assessing individual cells in tissues have suggested some effect but the exact degree of impact has been challenging to ascertain...” I found this sentence a bit vague; what exactly is it that single cell studies have shown and what is the challenge that remains?

We’ve clarified the paragraph to more precisely summarize the reports from the papers we’re citing. We have also added a citation to a review from 2021 (PMID:33234351). This has been a controversial topic and our vagueness really was intended to reflect the true state of the literature as we perceive it. Nonetheless, we concur that providing a clear and explicit explanation is the preferable approach.

The updated text is:

“Monoallelism is interesting because it could reflect regulatory noise from differentiation that is propagated forward epigenetically [9]. While some studies have reported this inherited effect in cell lines [12], more recent work assessing individual cells in tissues has suggested some effect but the exact degree of impact has been challenging to ascertain [13] with studies variously suggesting abundant monoallelism without lineage-dependency [10] and a major role for cell intrinsic noise such as bursting [14].”

Results: As a general comment – I could not understand the results section and figures without

a prior thorough read of the methods section. The authors may want to consider improving the readability of the paper by including a bit more information about their methods in the results section and figure legends. As an example: the second line of results refers to stability of overall gene expression as shown in figure 1 D. Figure 1D shows a cluster plot for one time point (the legend reads:” (D) Transcriptional similarity is higher between siblings than across armadillo quadruplets. Heatmap of sample-sample correlations for the first time point with example sample-sample scatterplots”). Thus, stability does not refer to variation that is consistent over time? The legend of figure 1D mentions with example sample-sample scatterplots, but I could not find the example scatterplots in the figure?

We improved readability by expanding Figure legends and providing more information about methods throughout the results section.

Regarding Fig. 1D, the scatter plots mentioned in the legend were moved to Fig. S1. The legend has been updated accordingly. As the reviewer judiciously points out, the word “stability” was ill-chosen in the context of Fig. 1D (inherited from a previous version of the paper) and was replaced with the word “similarity”. The panel now points out that variations in expression profiles are subtle, making it difficult to identify stable signatures later in the paper.

The numbering of families (2x beginning with 12 or 15 and the remaining 3 with 16): does this have a meaning? E.g. does 16 refer to 3 times the same mother?

The numbering of families is related to the year of collection (2012, 2015, or 2016) from independent pregnant females (since breeding in captivity is very challenging). This is now clarified in the Methods: “The names of the armadillo quadruplets were chosen according to the year of capture of the pregnant mother (e.g., 16-XX for 2016)”.

Figure 1:

- The correlations between overall gene expression profiles are large between all samples; I suppose that this might be driven by the many transcripts with an expression level at the extremes (hardly expressed at all versus abundantly expressed) – though it would be nice to show an example of individual level data (a scatterplot). It is not clear to me if the correlations are computed based on all transcripts or on the most variable subset? If it is based on all detected transcripts, have the authors considered computing the correlations for genome-wide expression profiles after standardization (i.e. Z-score) of the expression levels? Perhaps this might lead to a better discrimination of quadruplets, in terms of the amount of within-quad variation.
- Figure 1D shows “lane”, could the legend explain what it represents (I did find the information in the methods section, but it would be nice to also include this information in the legend).

In the main figure, we included the correlation computed on all genes (which we clarified in the legend), because this is both intuitive and easiest to compare to other studies. Following the reviewer’s suggestion, we compiled in Figure S1 six panels that show example scatter plots, along with the effect of subsetting to the top 1000 highly variable genes (HVG) and z-scoring before computing correlations. The scatter plots illustrate how genes are tightly correlated within quads (located along the diagonal), even after HVG selection, and more loosely correlated across quads (spread away from the diagonal). While HVG selection and z-scoring change the scale on which correlations are compared (0.95-1 range for all genes, 0.7-1 range for HVG, ~0-

1 range for z-scores), the overall structure of the data remains the same: siblings within quads form strongly correlated 4x4 blocks, suggesting strong genetic control of expression.

We updated the legend of Figure 1 by adding the following sentence: “Colors indicate quadruplet identity, sex (red=female, blue=male) and sequencing lanes for each sample”.

“Because this occurs after armadillo splitting,” Could the authors provide references/further details for this statement? At which stage (i.e. how many cells) does the splitting occur? (see also question above about introduction).

We have added information about when the separate embryos are first observed in the Introduction section. Due to the uncertainty around when the separate lineages are formed, we have edited this sentence so it no longer states that XCI occurs after splitting. The edited version is “This timing suggests that stochastic canalized variation between siblings is set in distinct cell lineages before separate armadillo embryos are observed (Fig. 2C).”

Not incorrect but difficult to read: defining a ratio as a difference (line98)

Agreed, we changed the definition of allelic ratios to “the fraction of expression attributed to each allele.”

Figure 2. The graphics are fantastic, but the legend is too slim for me to understand what is shown exactly in each figure. Could the legend be expanded a little bit to describe more precisely what is shown in each panel?

We expanded and clarified the legends for all the figures.

Could the authors comment on the extent to which their results might be influenced by post-splitting de novo (DNA sequence) mutations? Is anything known about the de novo mutation rate in armadillos (or could this be discussed in the light of observations in other species, i.e., human identical twins)?

*While the rate of de novo mutations in armadillos is unknown, based on germline de novo mutation rates from 36 mammals and the length of the *Dasyurus novemcinctus* genome, we estimate between 19 to 21 mutations per generation (most of which will have no meaningful effect). This information has been added to our Results section to allow comparisons with our finding of 700 genes with strong allelic imbalance on average. Although de novo mutations can be informative lineage markers, they are too rare in mammals to influence our results, which is summarized in this added text:*

“This is much larger than an estimated germline de novo mutation rate of 19 to 21 per generation based on data from 36 mammals, consistent with our model that these allelic imbalances arise epigenetically, like X-inactivation.”

Additionally, we have added citations to two relevant studies of twins to our Discussion section.

Line 242: this model assumes random mating in the parental generation?

Correct, our model assumes random mating in the parental generation, and we now clarify this in our Methods section: “We assume random mating in the parental generation so that each allele is inherited independently with probability μ ...”

Discussion: the opening sentence (“In human studies, expression variability is typically thought of as being due to some combination of environmental and genetic factors, leaving —noise purely as a nuisance term”) does not do justice to the alternative explanations that have been offered, e.g. a third source that may consist of nonlinear epigenetic processes that can create variability at all phenotypical-somatic and behavioral-levels). I recognize that this statement is refined on the next page.

We agree that the opening sentence insufficiently acknowledges the many nuanced views that have contributed to modeling gene expression variability. We do think it is helpful to lay out the simplest view first for a broad audience. We have edited the sentence to make it clearer that this is a first simple model and we now follow it with a more nuanced view echoing the reviewer’s language. Our Discussion section now opens with:

‘In human studies, expression variability is most simply thought of as being due to some combination of environmental and genetic factors, leaving “noise” purely as a nuisance term. More complex models may include epigenetic associations of phenotypic and behavioral interactions [21], but canalized noise is still rarely considered (although see, for example: [22,23]).’

Methods: Is it possible that/if the authors have assessed if the different quadruplets are related to each other (i.e. either because their mothers are related or because their fathers are related), and if the authors might want to discuss this option in the light of the high overall between-quadruplet correlation of gene expression profiles and potential further impacts on results.

The armadillos were collected from the wild and the genetic similarity does not suggest any particular relatedness (e.g., unique SNPs is uniform and the SNP overlap between pairs in Figure S6A is uniform). The very high gene expression correlation is likelier to be due to a strong degree of environmental control (despite, e.g., infection) and very high quality data (including alignment to personalized genomes). Probably the most important point for our study is that the similarity of the armadillo quads is fairly uniform as well (all equally distant from one another) with no overlaps driven by any other factors we could identify. This is now stated in the Methods: “The analysis of the overall transcriptome and the top 1000 highly variable genes (Fig. 1D, Fig. S1) suggests that all quads are roughly equally distant from each other (no evidence for any clear grouping or driver of variability such as age or SNP overlap shown in Fig. S6)”.

“Most litters in the semi-outdoor pens shared the pen with another litter (2 litters per pen).” statement sounds a bit odd to me, given that there were only 5 litters in total. Would it make sense to say that there were 2 pens with 2 litters and one pen with 1 litter?-

We apologize for the confusion. The litters were housed in a large holding facility, and each pen may have included a litter from a different study. We have changed this sentence to mention this: “Most litters in the semi-outdoor pens shared the pen with another litter, either from this study or a separate one (2 litters per pen)”

“At four to five months of age, the animals were delivered to the National Hansen’s Disease Program (NHDP) facility in Baton Rouge, LA where they were placed in pairs in modified rabbit cages” were they placed in pairs from the same litter?

Yes, siblings were placed together, and we have edited this sentence to clarify this pairing: “At four to five months of age, the animals were delivered to the National Hansen’s Disease

Program (NHDP) facility in Baton Rouge, LA where siblings were placed in pairs in modified rabbit cages.

During the course of the year, the original two of the five sets of armadillo quadruplets were infected intravenously in the saphenous vein with 1×10^9 Mycobacterium leprae
**I assume that this was done as part of the aim of another study. Could this be clarified and could the authors comment if this might have influenced results of the current study?

You are correct, this was done as part of another study, and we have added text to the Introduction section to clarify this:

“While individuals within litters share an environment at any given time, it was not a fixed one, with one major source of variability being the infection of the armadillos with the leprosy bacterium toward the end of our study (as a primary purpose of the colony).”

Furthermore, in line with our response to Reviewer 1, we have added text that discusses how this relates to our results in both the Introduction and Discussion sections.

“Finally, we took the variance of the extremal estimated skew values to estimate the number of cells” Please define “extremal” (this term has not been introduced yet)

The sentence has been rewritten as: “Finally, we took the variance of the estimated unfolded skew values $\{1-f_i, f_i\}$ to estimate the number of cells (N) in the original starting pool”. We further introduced mathematical notations throughout the paragraph to reduce ambiguity.

Could the authors comment how the lack of a pre-existing armadillo genome assembly and gene functional annotations for the armadillo might have affected their results?

We were very aware of this challenge because the representation of genetic diversity in reference genomes is a research interest of ours (PMID:35256454). The lack of an armadillo reference genome ultimately helped us detect allele specific expression because it required us to sequence quad-specific genomes. We have incorporated additional text to the Discussion section that describes how this increases our data quality with the drawback of increased cost:

“Our experimental design was enabled by our use of armadillos, but leveraging this non-model organism does create challenges with respect to genome annotation. We resolved this by sequencing the individual genomes and assembling the X-chromosome; this yields much higher quality data but at a much higher expense.”

Furthermore, we now deliberate on the possible reasons for the limited functional overlap in ASE signatures across the quads, including reference bias and other contributing factors.

The X-inactivation analysis is not entirely clear to me. Does it combine information across all genes? And does it only use heterozygous genotypes? Is it based on the same information for all quadruplets (or is it also possible that different transcript variants are used to interrogate X-inactivation in quadruplet 1 versus quadruplet 2?)

Yes, we are interrogating X-inactivation using all high-coverage heterozygous SNPs for a given quad. These SNPs may or may not be used in other quads because we aligned to quad-specific personalized genomes. While the transcript variants used are not identical, we only compare quads at a combined level by obtaining robust XCI ratios from all high-coverage SNPs.

We apologize for the confusion and have clarified the analysis throughout the results section and figure legends to clearly state how the various quads were used. For example, we now state that “To obtain allelic ratios, we aligned RNA sequencing reads to quad-specific personalized genomes, identifying a total of 26,325 heterozygous SNPs across the 5 quads.” to clarify that the analysis is based on heterozygous SNPs across all available quads, but that the SNPs are quad-specific (personalized genomes).

Please note that the GEO accession number was missing – perhaps this was not available yet. “The accession number for the sequencing datasets reported in this paper have been deposited in GEO under XXX”

Thank you for noticing this, we have deposited the data, and the GEO accession number was added: “GSE141951”. The dataset will be made public upon acceptance.

Reviewer #3

This is a laudible attempt to identify persistent allele-specific expression through the life of an animal. The mechanism could explain differences in penetrance of heterozygous rare variants. This is also modelled in the paper. A weakness of the paper is that the paper is purely based on RNA-seq data, and has not assessed DNA methylation levels (through genome-wide or targeted approaches).

Main comments

It is important to stress from the start that RNA sequencing was done on PBMCs so that the stochastic variation observed is the variation observed in blood cells and may not reflect that of other tissues. Moreover, the estimate of the precursor pool of cells (25 plus or minus one cell division for X-inactivation) is the precursor pool of blood cells, and should not be confused with the number of cells present in the embryo at the time of X-inactivation.

We thank the reviewer for their comments. In both the abstract and Introduction sections, we have added text to ensure the readers know we sampled blood. In our Discussion section, we now comment on the advantages and limitations of using PBMCs. Our cross-tissue analysis of human samples found that XCI ratios are generally shared across all tissues for an individual. This human data suggests that XCI is completed prior to germ-layer specification, and we are estimating the initial cell population when XCI occurs (PMID:35914524). As the reviewer points out, without testing another tissue, we cannot be certain and state that the generalizability of our results is limited in our Discussion section: “Of course, the use of PBMCs potentially limits the generality of the results.”

The authors should discuss the relation to published work in discordant monozygotic twins, e.g. a study on combined allele-specific methylation and gene expression, e.g.: <https://www.nature.com/articles/s41380-021-01126-w>; <https://www.ncbi.nlm.nih.gov/pmc/articles/PMC6887657/>

We have included these important citations to our Discussion section. However, many aspects of our study are innovative, such as the use of armadillos and the bioinformatic analyses we performed. While we sought to clarify aspects of the existing literature, our edits mainly focused on highlighting our contributions and improving the clarity and flow of our arguments.

The legends to the figures should contain much more detail. The authors should indicate what is plotted, based on how many SNPs and samples, what are the assumptions in the model etc.

We agree and have substantially expanded all figure legends.

The abstract needs to be rewritten to better explain the main points of the paper to a general audience. For example: the abstract starts with genetic and environmental variation. This is not what the paper is about. The paper is about the stochastic variation in allele-specific expression that persists through mitotic cell divisions. The term “developmental stochasticity” in the second sentence can still mean anything and the reader does not automatically associate this with gene expression. Make clear that this is about imbalanced expression from the two chromosomes. Also the implications for non- or variable penetrance and how this was modelled in the paper can be explained much more clearly.

We agree with the reviewer that our abstract was overly broad. We have rewritten it almost entirely to better capture our more specific points and focus. We did leave the initial sentence intact because we felt it started from a more conventional common starting perspective (GxE) in order to highlight how our focus differed from it. We specifically included the points mentioned by the reviewer, so it's now clear that we assayed blood samples from quadruplets of genetically identical individuals.

Minor comments

Make more clear that (if understood correctly) the quads are monozygotic and not heterozygotic, so are genetically identical? The value of the use of an animal model with four monozygotic quads can become clear in the abstract already.

We agree, and now include the phrase ‘genetically identical individuals’ in the abstract to emphasize that we profile monozygotic pups. We have also added more background information about armadillos in the Introduction section.

Even when using personal genomes, reference bias in the alignment may still persist. Describe the procedures to check for the absence of reference bias.

We think this is a very good point. In most of the paper, we avoid using absolute estimates of skews and generally focus on variability across individuals, which is only modestly impacted by reference bias. However, the timing of XCI (10-100 cells) relies on absolute skew estimation and is directly impacted by reference bias. To estimate reference bias and show the impact of reference bias on the computation of XCI, we added an additional analysis illustrated in Figure S2. In this figure, we take advantage of the fact that two quads have roughly opposite XCI ratios (deviating from 0.5 in opposite directions) to disentangle reference bias from true XCI skew. We estimate that the reference bias is around 0.034 and is unlikely to exceed 0.05. Then, we use simulations to estimate how reference bias impacts XCI timing estimation. We show that reference bias usually leads to an underestimation of XCI timing, but that, for the bias observed in the data and, given the early timing of XCI, our original estimation remains valid.

This additional analysis is now reported in the Results section: “After adjusting for reference bias, our estimate remains in a range of 10-100 cells (Fig. S2).”

The Methods section “X-chromosome inactivation analysis and cell number estimates” would benefit from more formula annotations. Currently, descriptions like “These gene ratios”, “extremal estimated skew values” are difficult to interpret.

We agree and have added formal definitions and formula annotations throughout the Methods section.

REVIEWERS' COMMENTS

Reviewer #3 (Remarks to the Author):

The authors have done a great job responding to my comments and those of the other reviewers.

Some minor remaining points:

- tenses are not always used consistently, please check
- some legends can still be improved. For many plots it is actually not clear which measure is plotted on the y-axis (e.g. 2D and others)
- For GEO entry, i would expect an GDS identifier that includes multiple samples (GSE identifiers)
- I suggest to be more cautious in extrapolating between species in the Discussion. For example, timing of XCI may differ between species

Reviewer #4 (Remarks to the Author):

We are satisfied with the careful way the reviewers' comments were addressed.

We thank the reviewers for their positive comments. Minor remaining tweaks are described below.

Review Comments:

The authors have done a great job responding to my comments and those of the other reviewers.

Some minor remaining points:

-tenses are not always used consistently, please check

We have checked the text and corrected tenses in multiple sentences.

-some legends can still be improved. For many plots it is actually not clear which measure is plotted on the y-axis (e.g. 2D and others)

We have add details to our figure legends to specify which measures are plotted.

-For GEO entry, i would expect an GDS identifier that includes multiple samples (GSE identifiers)

In GEO, GSM identifiers represent individual samples, and both the suggested GDS and the GSE identifier we use represent multiple samples. We provided a GSE identifier to link both the DNA and RNA sequencing data from this project.

-I suggest to be more cautious in extrapolating between species in the Discussion. For example, timing of XCI may differ between species

We have edited the concluding sentences of two paragraphs in the Discussion section to be more cautious in this regard.